# Pretreatment Masseter Muscle Volume Predicts Survival in Locally Advanced Nasopharyngeal Carcinoma Patients Treated with Concurrent Chemoradiotherapy

**DOI:** 10.3390/jcm12216863

**Published:** 2023-10-30

**Authors:** Umur Anil Pehlivan, Efsun Somay, Busra Yilmaz, Ali Ayberk Besen, Huseyin Mertsoylu, Ugur Selek, Erkan Topkan

**Affiliations:** 1Department of Radiology, Adana Dr. Turgut Noyan Application and Research Center, Faculty of Medicine, Baskent University, Adana 01120, Turkey; 2Department of Oral and Maxillofacial Surgery, Adana Dr. Turgut Noyan Application and Research Center, Faculty of Dentistry, Baskent University, Adana 01120, Turkey; efsuner@gmail.com; 3Department of Oral and Maxillofacial Radiology, School of Dental Medicine, Bahcesehir University, Istanbul 34349, Turkey; busra.yilmaz1@bau.edu.tr; 4Department of Medical Oncology, Medical Park Seyhan Hospital, Adana 07160, Turkey; besenay@gmail.com; 5Department of Medical Oncology, Medical Park Adana Hospital, Istinye University, Istanbul 34010, Turkey; mertsoylu@hotmail.com; 6Department of Radiation Oncology, Koc University School of Medicine, Istanbul 34010, Turkey; ugurselek@yahoo.com; 7Department of Radiation Oncology, Adana Dr. Turgut Noyan Application and Research Center, Faculty of Medicine, Baskent University, Adana 01120, Turkey; docdretopkan@gmail.com

**Keywords:** muscle loss, masseter muscle, locally advanced nasopharyngeal cancer, chemoradiotherapy, survival

## Abstract

Background and purpose: Muscle loss is a significant indicator of cancer cachexia and is associated with a poor prognosis in cancer patients. Given the absence of comparable studies, the current retrospective study sought to examine the correlation between the total masseter muscle volume (TMMV) before treatment and the survival outcomes in locally advanced nasopharyngeal cancer (LA-NPC) patients who received definitive concurrent chemoradiotherapy (CCRT). Methods: A three-dimensional segmentation model was used to determine the TMMV for each patient by analyzing pre-CCRT magnetic resonance imaging. The optimal TMMV cutoff values were searched using receiver operating characteristic (ROC) curve analyses. The primary and secondary endpoints were the relationship between the pre-CCRT TMMV measures and overall survival (OS) and progression-free survival (PFS), respectively. Results: Ninety-seven patients were included in this study. ROC curve analyses revealed 38.0 cc as the optimal TMMV cutoff: ≤38.00 cc (*n* = 42) and >38.0 cc (*n* = 55). Comparisons between the two groups showed that the TMMV>38.0 cc group had significantly longer PFS [Not reached (NR) vs. 28; *p* < 0.01] and OS (NR vs. 71; *p* < 0.01) times, respectively. The results of the multivariate analysis demonstrated that the T-stage, N-stage, number of concurrent chemotherapy cycles, and TMMV were independent associates of PFS (*p* < 0.05 for each) and OS (*p* < 0.05 for each) outcomes, respectively. Conclusion: The findings of the current retrospective research suggest that pretreatment TMMV is a promising indicator for predicting survival outcomes in LA-NPC patients receiving definitive CCRT.

## 1. Introduction

Nasopharyngeal carcinoma (NPC) is a distinct subset of head and neck cancers (HNCs) characterized by a marked degree of heterogeneity due to the anatomical locations from which it originates. It has a high incidence and prevalence, particularly in Far Eastern nations [1,2]. Although it has some limitations, the existing tumor-node-metastasis (TNM) staging system from the International Union Against Cancer (UICC) and American Joint Committee on Cancer (AJCC) is the most reliable prognostic tool for these patients [3]. Approximately 75% of all NPC patients present with a locally advanced NPC (LA-NPC) [4,5]. The current gold standard treatment for LA-NPCs is concurrent chemoradiotherapy (CCRT) with intensity-modulated radiation (IMRT), which has been shown to achieve local control rates of over 90% [3,6]. Notwithstanding the application of identical treatment protocols to patients with equivalent performance scales and TNM stages, disparate clinical outcomes may ensue. The existing TNM staging system weighs primary tumor extension, nodal dissemination, and distant metastasis. However, it neglects to account for pertinent biological parameters related to the patient and the disease, such as immune-inflammatory processes and nutritional status. This deficiency may be one of the primary reasons why diverse clinical outcomes arise after practically similar treatment procedures, underscoring the need for innovative biological indicators that may increase TNM's prognostic capacity.

Cancer cachexia is among the leading causes of cancer-related deaths. It is characterized by weight loss, along with a loss of muscle (sarcopenia) and/or fat mass [7]. Cancer cachexia ultimately results in compromised patient functionality and physical aptitude and is frequently linked to an unfavorable patient prognosis. Sarcopenia can be classified into two categories: primary sarcopenia, which is attributed to the natural process of aging, and secondary sarcopenia, which is caused by the presence of cytokines resulting from a range of pathologies, including advanced organ failure, chronic inflammatory diseases, endocrine diseases, and cancer [7]. Therefore, sarcopenia may serve as an efficient biological indicator to forecast cancer prognosis. Imaging modalities, such as computed tomography (CT) and magnetic resonance imaging (MRI), are frequently employed for the detection of sarcopenia [4,8]. Furthermore, the integration of image-based analyses with diverse biochemical and immunological parameters can be employed for the diagnosis of sarcopenia [7]. The assessment of sarcopenia in HNC through imaging-based investigations involves the utilization of the skeletal muscle index. This index is determined by dividing the cross-sectional area of skeletal muscle at either the C3 or L3 level [9,10] by the square of the individual's height in meters [1,4,8]. Likewise, the masseter muscle, which is one of the muscles involved in the mastication process, has garnered increased attention in the literature for the assessment of sarcopenia. The masseter muscle can be easily visualized and measured during routine imaging evaluations of HNC patients without necessitating an additional imaging protocol. Previous studies have consistently identified masseter muscle-defined sarcopenia as an unfavorable prognosticator for various cancers, including glioblastoma multiforme, esophageal cancer, and HNCs [11,12,13,14]. A correlation between the chosen muscle area or index and survival outcomes has been established in such studies [4,7,8,9,10,11,14]. The studies in question employed two-dimensional (2D) muscle areas as opposed to 3D measurements, which could potentially exhibit notable discrepancies among patients who possess nearly identical muscle areas. A recent study investigated the association between the masseter muscle volume and the overall survival (OS) in patients with HNCs [14]. The study found that although a reduction in the masseter muscle area and masseter skeletal muscle index (SMI) was linked with poor OS, there was no significant correlation between masseter muscle volume and OS [14].

To date, there has been no investigation into the possible correlation between the volume of the masseter muscle prior to treatment and the survival of LA-NPCs. Hence, the present retrospective study aimed to investigate the potential relationship between pretreatment total masseter muscle volume (TMMV) and survival outcomes in patients with LA-NPC undergoing definitive CCRT.

## 2. Materials and Methods

### 2.1. Ethics, Consent, and Permissions

The present investigation is a retrospective study conducted at a single center and adheres to the ethical standards set forth by the Institutional Clinical Research Ethical Committee as well as the Helsinki Declaration and its subsequent amendments. The study has received approval from the Baskent University Institutional Review Board under Project No. KA23/196. Before the commencement of the CCRT, all participants were required to provide written informed consent for the data collection and publication of related outcomes, per our institutional standards.

### 2.2. Patient Selection

A retrospective analysis was conducted on patients diagnosed with NPC who received CCRT at the Radiation Oncology Department of Baskent University Adana Application and Research Center during the period spanning from January 2010 to December 2022. The study's inclusion criteria comprised individuals who were 18 years of age or older, ECOG 0-1, body mass index (BMI) ≥ 18.5 kg/m^2^, had a confirmed diagnosis of nasopharyngeal squamous cell cancer through histopathological analysis, underwent neck MRI evaluation before any treatment, and were staged as LA-NPC (T_1–2_N_1–3_M_0_ or T_3–4_N_0–3_M_0_) according to the eighth edition of the AJCC staging framework [15]. The study excluded patients who did not have a follow-up period of at least six months, those who had facial asymmetry or dental malocclusion before CCRT, those who did not meet the eligibility criteria for treatment, those who had received treatment for a local recurrence, and those who had previously undergone surgery and/or radiotherapy in the head and neck region.

### 2.3. Masseter Muscle Volume Measurements

In all patients, a 20-channel head and neck coil, and either the MAGNETOM Skyra 3T or MAGNETOM Avanto Fit 1.5T, both manufactured by Siemens Healthcare in Erlangen, Germany, were used to obtain a neck MRI. The masseter muscle was evaluated using pre-contrast T2 weighted images without fat saturation. Parameters for pre-contrast T2 weighted images without fat saturation were as follows: TR: 7060 ms; TE: 79 ms; slice thickness: 3 mm; matrix: 224 × 320; field of view (FOV): 220 × 220 mm; number of excitations: 1; flip angle: 180 at 3T MR service and TR: 5550 ms; TE: 102 ms; slice thickness: 3 mm; matrix: 217 × 320; FOV: 230 × 230 mm; number of excitations: 2; flip angle: 150 at 1.5T MR service. The masseter muscle is a 3D tissue that originates from the zygomatic arch and inserts into the mandibular angle. A radiologist who specialized in head and neck imaging conducted volumetric analyses while blinded to the clinical data. Free-hand segmentation was performed using MIM 4.6.4 software (MIM Inc., Cleveland, OH, USA) to measure TMMVs (Figure 1).

### 2.4. Chemoradiotherapy Protocol

All the patients received definitive CRT. RT was administered daily, five days a week, for seven weeks, regardless of the technique used. The chemotherapy treatment plan consisted of cisplatin administered at a dosage of 75–80 mg/m^2^ every three weeks for the CCRT phase and two cycles of cisplatin-based regimens every three weeks for the adjuvant treatment phase.

### 2.5. Response Assessment and Follow-Up

Patients underwent weekly assessments or more frequent evaluations during the CCRT course, as deemed necessary. Following the conclusion of CCRT, patients underwent routine evaluations at three-month intervals during the initial two-year period, six-month intervals from the third to fifth years, and annually thereafter (or more frequently as required). At each visit, a comprehensive endoscopic head and neck examination was performed to detect potential local or regional tumor recurrences and assess the probability of second primary tumors. Furthermore, Positron Emission Tomography (PET)-CT scans were obtained to evaluate the emergence of distant metastasis and identify the appropriate treatment response. The standard tool used for response evaluations was the PET Response Criteria for Solid Tumors. PET-CT scanning was replaced with head and neck MRI and/or CT scans once a complete metabolic response was confirmed. Additional imaging techniques were used only to identify potential lesions or to reassess recurring tumors.

### 2.6. Statistical Analysis

The primary endpoint of this retrospective cohort study was the association between pre-CCRT TMMVs and OS: the interval between the onset of CCRT and death/last follow-up. The secondary endpoint was the association between pre-CCRT TMMVs and progression-free survival (PFS): the interval between the initiation of CCRT and any locoregional or distant disease progression or death/last follow-up. Continuous variables were described with medians and ranges, while categorical variables were conveyed with frequency distributions based on percentages. The patient groups were compared using the Chi-square test, Student’s t-test, and the Spearman correlation analysis, as indicated. Using receiver operating characteristic (ROC) curve analysis, the pre-CCRT TMMV cutoff that, if present, may divide the whole research cohort into two TMMV groups with different results was estimated. Similarly, we applied the same methodology to identify relevant cutoffs for other continuous variables, such as the patient's age or body mass index (BMI). The study employed univariate analyses to examine the potential interactions among patient, disease, and treatment variables concerning OS and PFS. Only factors that exhibited statistical significance were incorporated into the multivariate Cox proportional hazard model to ascertain their independent significance status. The statistical tests conducted were two-tailed and the level of statistical significance was set at *p* < 0.05.

## 3. Results

The clinical characteristics of 97 LA-NPC patients, consisting of 28 females (28.86%) and 69 males (71.14%), are summarized in Table 1. The median age was 57 (range: 19–79), and the majority of patients either had a T_1–2_ (64.94%) or N_2–3_ (63.91%) disease. The patients received treatment utilizing the IMRT technique. Concurrent chemotherapy was administered to 81 patients (83.5%), whereas 74 patients (76.3%) were able to receive adjuvant chemotherapy for 1–2 cycles.

The median follow-up time for the entire study cohort was 63 months (range, 6 to 141 months). Although the median OS duration was not reached during this final analysis, the median PFS was found to be 74 months (range, 32.1 to 115.9 months). During this analysis, 69 (71.1%) of the 97 patients were still alive, with 47 (48.5%) being free of disease progression. The 5-year PFS and OS rates were 54.3% and 78.7, respectively. The corresponding rates for 8-year PFS and OS were 44.7% and 63.9%, respectively.

The optimal TMMV cutoffs were 36.4 cc [Area under the curve (AUC): 71.4%; sensitivity: 75.4%; and specificity: 71.4%; Youden index: 0.468] for OS and 40.0 cc (AUC: 67.3%; sensitivity: 63.8%; and specificity: 62.0%; Youden index: 0.258] for PFS (Figure 2). The cutoff point of 38 cc was established as the standard for TMMV for ease of further analysis, as there were only two discrepancies observed between patients at this cutoff point. Hence, the patients were categorized into two groups based on this cutoff value: TMMV ≤ 38.0 cc (*n* = 42) and TMMV > 38.0 cc (*n* = 55). There were no significant differences between the two groups in terms of age, gender, ECOG performance status, BMI, T-stage, N-stage, number of concurrent chemotherapy cycles, and number of adjuvant chemotherapy cycles (Table 1). The median PFS and OS were 28 and 71 months for TMMV ≤ 38.0 cc group, while neither of them was reached yet for the TMMV > 38.0 cc group during the final analysis. The difference between the two groups was significant for median PFS (*p* < 0.001) and OS (*p* < 0.001) durations (Figure 3). Similarly, higher 5-year and 8-year PFS (34.4% vs. 69.1%, and 22.9% vs. 59.6%) and OS (59.1% vs. 93.9%, and 37.1% vs. 83.0%) rates were associated with higher pretreatment TMMV values (Table 2).

Univariate analysis revealed that T_1–2_ tumor stage, N_0–1_ nodal stage, administration of 2–3 concurrent chemotherapy cycles, and TMMV > 38 cc were associated with significantly superior OS and PFS outcomes (Table 3). The results of the multivariate Cox regression analysis indicated that T_1–2_ stage [Hazard ratio (HR): 1.34, *p* = 0.027 for PFS; HR: 1.38, *p* = 0.022 for OS], N_0–1_ stage (HR: 1.36, *p* = 0.014 for PFS; HR: 1.43, *p* = 0.029 for OS), administration of 2–3 concurrent chemotherapy cycles (HR: 1.42, *p* = 0.043 for PFS; HR: 1.73, *p* = 0.032 for OS), and TMMV > 38 cc (HR: 2.38, *p* < 0.001 for PFS; HR: 2.34, *p* < 0.001 for OS) could independently predict better OS and PFS (Table 3).

## 4. Discussion

The main objective of this research was to examine whether there was a correlation between pretreatment TMMV values and the survival results of LA-NPC patients following CCRT. According to our findings, a lower TMMV was a significant independent prognostic factor for worse PFS and OS outcomes in this particular patient group (*p* < 0.001).

In the last decade, several studies have explored the potential of using muscle loss, also known as sarcopenia, as a biomarker to predict survival in individuals with HNCs [1,4,8,11,16,17]. Consistent findings from these studies implied that severe muscle loss, whether present before treatment or caused by the treatment, is an adverse prognostic indicator for these patients. In their study, McGoldrick et al. and van Heusden et al. investigated the correlation between sarcopenia defined by masseter muscle area and survival in HNCs [11,14]. The findings of these studies have indicated that a smaller masseter muscle area is linked to inferior survival outcomes. Nevertheless, until the present study, there has never been a study that exclusively analyzed the volumetric data of the masseter muscle to characterize sarcopenia in LA-NPC patients. van Heusden et al. found a strong correlation between masseter SMI and volume, but only SMI could predict OS in HNC patients, with no significant relationship between masseter volume and OS [14]. But verification of these results through additional studies is necessary, since the third dimension may have equalized the masseter muscle volumes despite differences in the areas. We believe that volumetric analysis is a more precise method for evaluating sarcopenia due to the 3D nature of muscle tissues. To clarify, muscles possessing distinct third dimensions may be classified differently based on the variations in their third dimension despite having identical axial surface areas. To the best of our knowledge, our study is the first to assess the relationship between pretreatment TMMV values and survival outcomes in LA-NPC patients who underwent definitive CCRT.

In their study, Huang et al. reported that the occurrence of significant muscle loss after treatment serves as a prognostic indicator for OS in patients diagnosed with non-metastatic NPC. However, the study did not establish any correlation between lower muscle status before treatment and OS outcomes [1]. However, the study by Hua et al., which had a larger sample size, determined that pre-treatment lower muscle status was a robust predictor of worse PFS and OS in patients with LA-NPC [4]. Also, McGoldrick et al. reported that pre-treatment muscle loss in patients with HNCs was an independent predictor of poorer PFS and OS [11]. Consistent with the findings of Hua et al. and McGoldrick et al., our study reveals that low TMMV serves as a significant and independent biomarker that reliably predicts inferior median and long-term PFS and OS outcomes following CCRT in LA-NPC patients [4,11]. However, our study is different from these studies as it uses masseter muscle volumetric analysis, which we believe provides a more reliable assessment of muscle status.

The main finding of our study was the first-time demonstration of an independent and durable association between the pre-CCRT TMMV values and PFS and OS outcomes in LA-NPC patients. Additionally, the freehand segmentation volume analyses method appeared to be efficiently performed in daily routine for patients enrolled in the IMRT program. Such study results may prove useful for more accurately predicting the treatment outcomes and tailoring their treatment. For example, early supportive care measures in high-risk patients may achieve better treatment tolerability and results. Our results also confirmed the T-stage, N-stage, and concurrent chemotherapy cycles as the other independent determinants of the survival results in this patient group. Therefore, it appears that patients who had a lower T-stage and/or N-stage before treatment and received a higher number of concurrent chemotherapy cycles had a better prognosis after CCRT. Similar results were also reported by Hua et al. in a previous study [4]. We believe that the correlation between the number of concurrent chemotherapy cycles and survival is linked to the patient's ability to tolerate the treatment. This belief has been supported by other HNC studies demonstrating a strong correlation between muscle loss and increased rates of treatment-related adverse effects, the need for dose reductions, and the inability to complete the prescribed treatment protocol [4,16,18].

Although we established notable correlations between TMMV and OS and PFS outcomes in our study, the precise mechanisms that account for this relationship remain unclear. However, it is plausible that the observed phenomenon can be attributed to the manifestation of cancer cachexia, a condition characterized by the depletion of both muscular and adipose tissue [7]. To enhance comprehension of this particular medical condition, it is of utmost importance to assess the correlation between proinflammatory mediators, namely interleukins and tumor necrosis factor-alpha (TNF-α), and muscle loss. The occurrence of cancer-related muscle loss is profoundly influenced by cytokine-mediated inflammation, specifically involving TNF-α, interleukin-1 (IL-1), IL-6, and interferon-gamma (IFNγ) [19,20,21]. The activation of the nuclear factor kappa B (NF-kB) pathway by TNF-α can result in the degradation of muscle proteins via ubiquitin-mediated proteasome and the upregulation of uncoupling proteins (UCPs) 2 and 3, ultimately leading to muscle catabolism [19,20,21]. The exact correlation between IFNγ and cachexia remains ambiguous; however, research has demonstrated its potential to collaborate with TNF-α in this regard [21]. The NF-KB pathway is thought to be the mechanism through which muscle loss is initiated by IL-1 [20]. Furthermore, IL-6 has been linked to the initiation of catabolic mechanisms by inhibiting protein synthesis via the muscle Janus kinase signaling pathway and increasing the expression of UCP1 [20]. Hence, a more in-depth investigation into the function of cytokine-induced inflammation and cancer-associated muscle atrophy, with a particular focus on the participation of TNF-α, IL-1, IL-6, and IFNγ, could yield significant knowledge regarding the fundamental mechanisms involved.

The present investigation is subject to various constraints. First, it has a single-centered retrospective design with a relatively limited cohort size. Second, while previous muscle area-based studies have utilized distinct cutoff values for males and females [1,8,11], we opted for a unified cutoff value for both genders. Third, our study did not assess the likelihood that low TMMV was initially a component of cancer cachexia. Hence, the correlation between TMMV and established predictive biomarkers of cancer cachexia, including IL-1, TNF-alpha, IL-6, CRP, and serum albumin, warrants investigation. And fourth, salvage treatments may have varied among patients, which could have inadvertently resulted in a bias towards one TMMV group over the other with respect to survival outcomes. Consequently, it is advisable to exercise prudence when interpreting the present findings and to regard them as hypothetical rather than conclusive recommendations until subsequent research, employing suitable methodology and involving substantial sample sizes, corroborates them.

## 5. Conclusions

In conclusion, this study represents the first evidence demonstrating the relationship between higher pretreatment TMMV measures and superior survival outcomes in LA-NPC patients treated with definitive CCRT. If substantiated by additional research, it could be possible to utilize imaging modalities obtained before treatment initiation to tailor treatment and provide personalized supportive care to these patients.

## Figures and Tables

**Figure 1 jcm-12-06863-f001:**
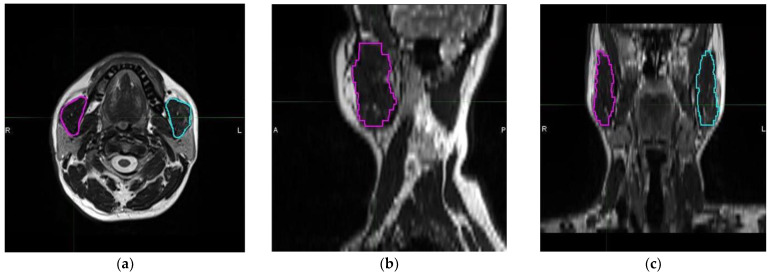
Representative delineation and measurement process of each masseter muscle using T2 weighted magnetic resonance imaging scans: (**a**) axial, (**b**) sagittal, and (**c**) coronal planes (magenta: right masseter muscle; cyan: left masseter muscle). (R: Right, L: Left, A: Anterior, and P: Posterior).

**Figure 2 jcm-12-06863-f002:**
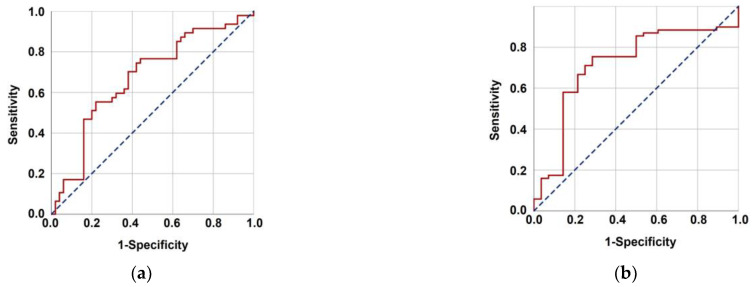
Results of the characteristics receiver operating characteristic curve analysis evaluating the relationship between pretreatment total masseter muscle volume with progression-free survival (**a**) (area under the curve (AUC): 67.3%; sensitivity: 63.8%, and specificity: 62.0%; Youden index: 0.258) and overall survival (**b**) (AUC: 71.4%; sensitivity: 75.4%, and specificity: 71.4%; Youden index: 0.468).

**Figure 3 jcm-12-06863-f003:**
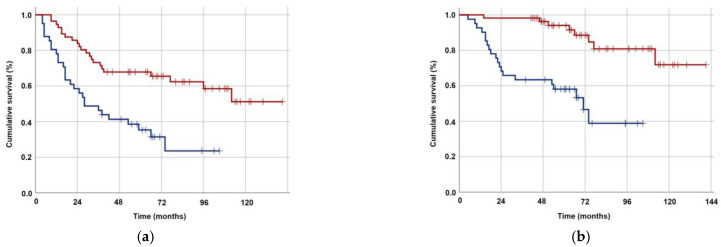
Kaplan–Meier survival curves for groups with TMMV ≤38.0 cc (dark blue) versus >38 cc (red). (**a**) Progression-free survival. (**b**) Overall survival.

**Table 1 jcm-12-06863-t001:** Baseline and treatment characteristics for all patients and per total masseter volume groups.

Characteristics	All Patients	TMMV ≤ 38.0 cc	TMMV > 38.0 cc	*p*-Value
	(*n* = 97)	(*n* = 42)	(*n* = 55)	
Median age, year (range)	57 (19–79)	56 (28–76)	57 (19–79)	0.78
Age group, *n* (%)				
≥65 years	38 (39.17%)	17 (40.48%)	21 (38.18%)	0.43
<65 years	59 (60.83%)	25 (59.52%)	34 (61.82%)	
Gender, *n* (%)				
Female	28 (28.86%)	13 (30.96%)	15 (27.27%)	0.51
Male	69 (71.14%)	29 (69.04%)	40 (72.73%)	
ECOG, *n* (%)				
0	41 (42.26%)	19 (45.24%)	22 (40%)	0.82
1	56 (57.74%)	23 (54.76%)	33 (60%)	
Median BMI, kg/m^2^	22.7 (18.6–26.8)	22.2 (18.6–25.8)	23.3 (18.8–26.8)	0.29
BMI group, *n* (%)				
<22.7 kg/m^2^	41 (42.2)	18 (42.6)	23 (41.8)	0.72
≥22.7 kg/m^2^	56 (57.8)	24 (57.4)	32 (58.2)	
T-stage *, *n* (%)				
1–2	63 (64.94%)	31 (73.81%)	32 (58.18%)	0.13
3–4	34 (35.06%)	11 (26.19%)	23 (41.82%)	
N-stage *, *n* (%)				
0–1	35 (36.09%)	17 (40.48%)	18 (32.73%)	0.52
2–3	62 (63.91%)	25 (59.52%)	37 (67.27%)	
Concurrent chemotherapy cycles, *n* (%)				
1	16 (16.50%)	7 (16.66%)	9 (16.36%)	0.86
2–3	81 (83.50%)	35 (83.34%)	46 (83.64%)	
Adjuvant chemotherapy cycles *n* (%)				
0	23 (23.7%)	11 (26.20%)	12 (21.82%)	0.48
1–2	74 (76.3%)	31 (73.80%)	43 (78.18%)	

* According to the eighth edition of the AJCC. Abbreviations: TMMV: total masseter muscle volume; ECOG: Eastern Cooperative Oncology Group; BMI: Body mass index; T-stage: Tumor stage; N-stage: Nodal stage.

**Table 2 jcm-12-06863-t002:** Median and long-term survival outcomes for the entire research cohort and per total masseter muscle volume groups.

Endpoint	All Patients (*n* = 97)	TMMV ≤ 38.0 cc (*n* = 42)	TMMV > 38.0 cc (*n* = 55)	*p*-Value
PFS				
Median, months	74 (32.1–115.9)	28 (13.3–42.7)	NR	<0.001
5-year survival, %	54.3	34.4	69.1	
8-year survival, %	44.7	22.9	59.6	
OS				
Median, months	NR	71 (57.3–84.7)	NR	<0.001
5-year survival, %	78.7	59.1	93.9	
8-year survival, %	63.9	37.1	83	

Abbreviations: TMMV: total masseter muscle volume; PFS: progression-free survival; NR: not reached; OS: overall survival.

**Table 3 jcm-12-06863-t003:** Outcomes of univariate and multivariate analyses.

Variable	Progression-Free Survival				Overall Survival			
	Median survival (months)	Univariate *p*-value	Multivariate *p*-value	HR (95%CI)	Median survival (months)	Univariate *p*-value	Multivariate *p*-value	HR
								(95%CI)
Age group								
<65 years	NR	0.52	-	-	NR	0.62	-	-
≥65 years	NR				NR			
Gender								
Female	NR	0.74	-	-	NR	0.58	-	-
Male	NR				NR			
ECOG								
0	NR	0.82	-	-	NR	0.71	-	-
1	NR				NR			
BMI group, *n* (%)								
<22.7 kg/m^2^	NR	0.91	-	-	NR	0.94	-	-
≥22.7 kg/m^2^	NR				NR			
T-stage *								
1–2	NR	0.012	0.027	1.34 (1.17–1.58)	NR	0.011	0.022	1.38 (1.21–1.59)
3–4	66				NR			
N-stage *								
0–1	NR	0.023	0.014	1.36 (1.13–1.68)	NR 112	0.014	0.029	1.43 (1.18–1.74)
2–3	57							
Concurrent chemotherapy cycles								
1	53	0.032	0.043	1.42 (1.14–1.86)	71	0.012	0.032	1.73 (1.26–2.24)
2–3	NR				NR			
Adjuvant chemotherapy cycles								
0	NR	0.64	-	-	NR	0.38	-	-
1–2	NR				NR			
TMMV								
≤38.0 cc	28	<0.001	<0.001	2.38 (1.82–2.94)	71	<0.001	<0.001	2.34 (1.67–3.01)
>38.0 cc	NR				NR			

* According to the eighth edition of the AJCC. Abbreviations: HR: hazard ratio; CI: confidence interval; NR: not reached; ECOG: Eastern Cooperative Oncology; BMI: body mass index; T-stage: tumor stage; N-stage: node stage; TMMV: total masseter muscle volume.

## Data Availability

The present data belong to and are stored at the Baskent University Faculty of Medicine and cannot be shared without permission. For researchers who meet the following criteria for access to confidential data, contact the Baskent University Corporate Data Access/Ethics Board: adanabaskent@baskent.edu.tr.

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
