# Peer review of "Pretreatment Masseter Muscle Volume Predicts Survival in Locally Advanced Nasopharyngeal Carcinoma Patients Treated with Concurrent Chemoradiotherapy"

_jcm, 2023, doi:10.3390/jcm12216863_

Round 1

Reviewer 1 Report

Comments and Suggestions for Authors

The authors showed that there was a correlation between masseter muscle volume and treatment outcomes in patients with nasopharyngeal cancer receiving chemoradiotherapy. Using a threshold of 38 cc of masseter muscle volume generated by ROC curve analysis, patients with less volume were shown to have worse treatment outcomes in terms of overall survival and progression-free survival. Masseter muscle volume was a significant factor on both monovariate and multivariate analysis. 

This article is well written and of particular interest because predicting treatment outcomes for patients with nasopharyngeal cancer has been in great need, as the current TNM classification is not perfect for prognostication, as the authors mentioned.

However, several points should be corrected.

1) Should the volume of the masseter muscle be assessed in relation to the size of the patients, such as weight and body mass index? Consideration of relative muscle volume may be more appropriate than absolute muscle volume.   

2) Age is known to correlate with outcomes in patients with head and neck cancer. Age should be used as a continuous variable in univariate and multivariate analysis, rather than being divided into two groups: under 65 years or not. In Table 1, those in the older patient group have a numerically smaller masseter muscle compared to the other group, although this is not statistically significant. 

3) Last but not least, how would this new method of prognostication be useful in practice? Should patients be treated with dose escalation of de-escalation according to the result of the prediction? The authors should mention its usefulness in the discussion section.

Author Response

Dear Editor,

Thank you for your letter regarding our manuscript "Pretreatment Masseter Muscle Volume Predicts Survival in Locally Advanced Nasopharyngeal Carcinoma Patients Treated with Concurrent Chemoradiotherapy." We appreciate the comments from you and the reviewers. We are pleased to inform you that the manuscript has been subjected to a meticulous revision process, taking into account all the valuable feedback provided by the reviewers. We have carefully incorporated the suggested changes and addressed the concerns raised while ensuring that the overall coherence and clarity of the work are maintained. Please find a detailed account of the point-by-point modifications made to the manuscript enclosed with this letter. We hope our revisions are satisfactory and acceptable for further consideration for publication in the highly esteemed Journal of Clinical Medicine. All co-authors have read and approved the revised manuscript. We appreciate your time and effort in reviewing our work and look forward to hearing from you soon.

Thank you so much for your kind consideration!

Yours sincerely,

On behalf of all coauthors

Umur Anil Pehlivan, MD

Reviewer 1

We appreciate the valuable suggestions provided by Reviewer 1.

Comment 1: Should the volume of the masseter muscle be assessed in relation to the size of the patients, such as weight and body mass index? Consideration of relative muscle volume may be more appropriate than absolute muscle volume.

Response 1: In this study, all our patients had a BMI of 18.5 or higher. We have now included the relevant information about this topic in the methodology section of the report (Line 110). We have revised Table 1 to comparatively display the median BMI values for the two TMMV groups, as the ROC analyses did not reveal any meaningful cut-off value for BMI that divides patients into two distinct survival groups. Additionally, the outcomes of univariate and multivariate analysis based on BMI groups have been included in Table 3 and the statistical analysis section (Line 170-172).

Comment 2. Age is known to correlate with outcomes in patients with head and neck cancer. Age should be used as a continuous variable in univariate and multivariate analysis, rather than being divided into two groups: under 65 years or not. In Table 1, those in the older patient group have a numerically smaller masseter muscle compared to the other group, although this is not statistically significant. 

Response 2. A cut-off value for age was not determined through the ROC analysis. Due to this, we followed the classification by the World Health Organization and categorized the patients into two groups - one below the age of 65 (young) and the other above the age of 65 (elderly).

Comment 3. Last but not least, how would this new method of prognostication be useful in practice? Should patients be treated with dose escalation of de-escalation according to the result of the prediction? The authors should mention its usefulness in the discussion section.

Response 3. We want to express our gratitude for your comment, which will undoubtedly enhance the impact of our investigation. Dose escalation or de-escalation is a highly debated topic for any cancer group, particularly for head and neck cancers. However, it is challenging to recommend any of these options to nasopharyngeal cancer patients without a properly designed randomized trial addressing this issue as per TMMV groups.  However, such a novel prognostication may yield some clinical advantages, as discussed in the revised Discussion section. (Lines 297-303).

Reviewer 2 Report

Comments and Suggestions for Authors

This is an interesting study on the  prognostic value of masseter volume in nasopharyngeal carcinoma.

I think it is better to refer to original papers of L3 and C3 measurement techniques (in stead of (1,4,8): for example Shen et al. J Appl Physiol (1985). 2004 Dec;97(6):2333-8. and Swartz et al. Oral Oncol. 2016 Nov;62:28-33. 

Measuring masseter volume (3D) as prognostic factor in head and neck cancer has been done before. Please refer also to Van Heusden et al Quant Imaging Med Surg. 2022 Jan;12(1):15-27. Please add to the introduction. In this study also the correlation with CSMA of C3 and L3 was investigated. Please add.

Why was C3 not on the CT/MRI-scan routinely made for NPC? CSMA C3 has a better correlation with CSMA L3 the reference standard for total body muscle mass after correction for heigth.

Masseter muscle volume is likely to be influenced by dental state. Do you have information on this in your series? Please add to discussion.

Minor comments:

First sentence Discussion. I think that 'connection' is not the right word. 'correlation' is better.

Comments on the Quality of English Language

Well written

Author Response

Dear Editor,

Thank you for your letter regarding our manuscript "Pretreatment Masseter Muscle Volume Predicts Survival in Locally Advanced Nasopharyngeal Carcinoma Patients Treated with Concurrent Chemoradiotherapy." We appreciate the comments from you and the reviewers. We are pleased to inform you that the manuscript has been subjected to a meticulous revision process, taking into account all the valuable feedback provided by the reviewers. We have carefully incorporated the suggested changes and addressed the concerns raised while ensuring that the overall coherence and clarity of the work are maintained. Please find a detailed account of the point-by-point modifications made to the manuscript enclosed with this letter. We hope our revisions are satisfactory and acceptable for further consideration for publication in the highly esteemed Journal of Clinical Medicine. All co-authors have read and approved the revised manuscript. We appreciate your time and effort in reviewing our work and look forward to hearing from you soon.

Thank you so much for your kind consideration!

Yours sincerely,

On behalf of all coauthors

Umur Anil Pehlivan, MD

Reviewer 2

We are grateful for the valuable comments provided by Reviewer 2.

Comment 1. I think it is better to refer to original papers of L3 and C3 measurement techniques (in stead of (1,4,8): for example Shen et al. J Appl Physiol (1985). 2004 Dec;97(6):2333-8. and Swartz et al. Oral Oncol. 2016 Nov;62:28-33. 

Response 1. Thank you for your valuable suggestion. The purpose of this section was to define the skeletal mass index (SMI). However, we have taken into account the articles you mentioned which explain how area measurements are made, and we have added references to these articles in the relevant tab. We hope this clarifies any confusion and improves the overall quality of the section. However, we believe it would be convenient to retain the references to 1, 4, and 8 since more recent articles are available highlighting the subject of SMI, as stated in Lines 74-75.

Comment 2. Measuring masseter volume (3D) as prognostic factor in head and neck cancer has been done before. Please refer also to Van Heusden et al Quant Imaging Med Surg. 2022 Jan;12(1):15-27. Please add to the introduction. In this study also the correlation with CSMA of C3 and L3 was investigated. Please add.

Response 2. The revised Introduction and Discussion sections have added information and inferences about van Heusden et al.'s article. (Lines 85-90, 268-277).

Comment 3. Why was C3 not on the CT/MRI-scan routinely made for NPC? CSMA C3 has a better correlation with CSMA L3 the reference standard for total body muscle mass after correction for heigth.

Response 3. The C3 region is also scanned during routine MRI in such patients; however, the present study specifically aimed to evaluate the relationship between TMMV and OS. Hence, comparing their relative predictive powers may be the subject of another study. We believe that the masseter muscle is functionally more appropriate than the C3 skeletal muscles because it is the strongest muscle of mastication and is an essential indicator of the patient's nutritional status for assessing sarcopenia. Also, three-dimensional measurement is potentially more accurate than two-dimensional measurement, as the third dimension may change the measures even if the first two dimensions are identical. Hence, in Lines 83-85 of the text, we state: "The studies in question employed two-dimensional muscle areas as opposed to three-dimensional measurements, which could potentially exhibit notable discrepancies among patients who possess nearly identical muscle areas.". When assessing a patient with kyphosis or lordosis, the muscle area measurement may not accurately reflect its original value in a physiologic position due to the patient's positioning during imaging. Therefore, volumetric analysis is a more reliable indicator than area analysis.

Comment 4. Masseter muscle volume is likely to be influenced by dental state. Do you have information on this in your series? Please add to discussion.

Response 4. Thank you for reminding us of the importance of addressing dental malocclusion and facial asymmetry in our study. To prevent any potential biasing effects, we have excluded patients with these conditions prior to treatment and have now added them to the exclusion criteria section in the Methods of our revised manuscript, specifically on Line 114-118. We appreciate your valuable feedback.

Comment 5. First sentence Discussion. I think that 'connection' is not the right word. 'correlation' is better.

Response 5. According to your suggestions, "correlation" was replaced with "connection.." (Line 260).

Reviewer 3 Report

Comments and Suggestions for Authors

Dear Authors,

Thank You for submitting Your work, which I have read with great interest. I have a few comments/questions which I detailed here below and for which I am asking for a feedback from Your part:

- In the methods You describe that patients enrolled underwent concurrent chemoradiotherapy /CCRT), however not all subjects underwent the same treatment and some of them underwent chemotherapy as an adjuvant treatment after exclusive RT. Please give details in the methods section and inclusion criteria

- Please give more details and explain what You mean by: "While the median OS of all patients was not reacted" (line 184) "while neither of them was reached yet for the TMMV>38.0 " (line 198) and al the "not reached" in Table 3.

- I know it might sound obvious, but I think it is of utmost importance to specify the direction of the association between your independent variables and your outcomes. For example: does a hight T stage correlate with longer or shorter OS? And so on.... (I am particularly referring to the paragraph at page 7 in the results section, but this also applies to Table 3!)

Author Response

Dear Editor,

Thank you for your letter regarding our manuscript "Pretreatment Masseter Muscle Volume Predicts Survival in Locally Advanced Nasopharyngeal Carcinoma Patients Treated with Concurrent Chemoradiotherapy." We appreciate the comments from you and the reviewers. We are pleased to inform you that the manuscript has been subjected to a meticulous revision process, taking into account all the valuable feedback provided by the reviewers. We have carefully incorporated the suggested changes and addressed the concerns raised while ensuring that the overall coherence and clarity of the work are maintained. Please find a detailed account of the point-by-point modifications made to the manuscript enclosed with this letter. We hope our revisions are satisfactory and acceptable for further consideration for publication in the highly esteemed Journal of Clinical Medicine. All co-authors have read and approved the revised manuscript. We appreciate your time and effort in reviewing our work and look forward to hearing from you soon.

Thank you so much for your kind consideration!

Yours sincerely,

On behalf of all coauthors

Umur Anil Pehlivan, MD

Reviewer 3

We are grateful to Reviewer 3 for taking the time to provide us with his/her thoughtful and constructive suggestions. Their input has been incredibly helpful in improving our work.

Comment 1. In the methods You describe that patients enrolled underwent concurrent chemoradiotherapy /CCRT), however not all subjects underwent the same treatment and some of them underwent chemotherapy as an adjuvant treatment after exclusive RT. Please give details in the methods section and inclusion criteria

Response 1. By 'the same treatment,' we meant that all patients were prescribed the same chemotherapy and radiotherapy protocols. However, individual patient tolerances during the concomitant or adjuvant phases may vary, requiring adjustments. Furthermore, some patients may be less willing to continue the prescribed chemotherapy due to the toxicities associated with the treatment or their personal choice. It is not uncommon for patients to receive a chemotherapy cycle count that differs from their original prescription, which can pose a challenge in comparative analysis. However, there were no significant differences between the two TMMV groups regarding these factors (Table 1). As such, the probability of these factors impacting the findings presented in our article is statistically inconsequential.

Comment 2. Please give more details and explain what You mean by: "While the median OS of all patients was not reacted" (line 184) "while neither of them was reached yet for the TMMV>38.0 " (line 198) and al the "not reached" in Table 3.

Response 2. We used the phrase "not reached yet" to convey that the cumulative event occurrences, such as death or failure, depending on the endpoint, did not reach a statistically significant level. In other words, we used this expression to indicate that the median overall survival of the group had not been reached yet. However, we have revised the related sentences to clarify this issue, as can be seen in lines 190-195.

Comment 3. I know it might sound obvious, but I think it is of utmost importance to specify the direction of the association between your independent variables and your outcomes. For example: does a high T stage correlate with longer or shorter OS? And so on.... (I am particularly referring to the paragraph at page 7 in the results section, but this also applies to Table 3!)

Response 3. We have revised the necessary lines (233-240) to provide better clarity on this issue.

Round 2

Reviewer 1 Report

Comments and Suggestions for Authors

The authors corrected the manuscript properly according to the reviewer's comments.